# Geographic proximity, supply chain and organizational glocalized survival: China's e-commerce investments in Indonesia

**Jinsheng (Jason) Zhu**[1,2¤a¤b], **Weidian Lan**[3¤c], **Xianchun Zhang**[2¤b]*

**1** Faculty of Social Sciences, Chiang Mai University, Chiang Mai, Thailand, **2** Belt and Road International School, Guilin Tourism University, Guilin, Guangxi, China, **3** Faculty of Applied Linguistics, University of Warwick, Coventry, United Kingdom

¤a Current address: Chiang Mai University, Suthep, Chiang Mai, Thailand
¤b Current address: Belt and Road International School, Guilin Tourism University, Guilin, Guangxi, China
¤c Current address: University of Warwick, Coventry, United Kingdom
* zhang6022@126.com

**Data Availability Statement:** All relevant data are within the paper and its Supporting Information files.

## Abstract

Chinese e-commerce companies are in the ascendant into the overseas market, while still lack adequate academic attention. Adopting case study and public policy approaches, this article applies the symbiosis theory, based on the fundamentals of the development data of Chinese e-commerce companies in the Indonesia market, to construct an evaluation model and proposes a strategic orientation to reaching an embedded survival and further development. Through understanding the structural characteristics and developing status of different types of Chinese e-commerce companies going overseas, a detailed investigation to the Chinese e-commerce companies investing in Indonesia has been conducted. Findings show that the production capacity cooperation stage of the two countries has a trend of asymmetric symbiosis gradually developing towards symmetric symbiosis. To promote a continuous economic cooperation between China and Indonesia, this article proposes that the national-level collaboration policies, cross-border e-commerce value chain, as well as organizational-level coordination are the key sectors for reaching the vision of symmetric symbiosis between the two countries. Sectors in infrastructure, trade, capital, and people's mindset intimacy also contribute to construct a symbiosis mechanism for capacity cooperation between the two nations.

## Introduction

Cross-border e-commerce is set to grow immensely in Asia in the past decade (refer to Fig 1 below as international internet users increasingly high year on year) [1]. In recent years, China's economic and trade exchanges with developing countries such as Southeast Asia and South Asia have become increasingly important. The industrial structure has become increasingly integrated among the countries alongside the silk road. At the same time, China has become the world's biggest e-commerce market, neighbouring countries like Indonesia, Vietnam, and the Philippines grow at impressive rates. Indonesia, as one of the key players in

**Funding:** 1. National Office for Philosophy and Social Sciences 17BJY150 2. Guilin Tourism University KJ0603227

**Competing interests:** The authors have declared that no competing interests exist.

Source: ITU (International Telecommunication Union 2019)

**Fig 1. Individuals using the internet from 2005 until 2019.**

international trade to China, is a relatively underdeveloped country in Southeast Asia. Indonesia plays a more important role in China-Indonesia e-commerce cooperation and enterprise collaboration. Since 2016, China-Indonesia has maintained close economic and trade exchanges, and the bilateral e-commerce trade volume has increased year by year. By 2019, the bilateral international trade volume reached 79.7 billion US dollars, an annual increase of 3.1%. Among them, China's exports to Indonesia were US$45.64 billion, and imports from Indonesia were US$34.06 billion, an increase of 5.7% and a decrease of 0.3% respectively [2]. In Indonesia, Tokopedia had the first-mover advantage and now controls the biggest slice of the country's e-commerce market. It works by connecting buyers with sellers so there is no need for huge, capital-intensive logistics facilities, making the outfit attractive to investors.

China and Indonesia are undeniably playing essential roles in the cross-border e-commerce collaborations. In 2017, the president of China put forward the initiative of "discussing cooperation plans, jointly building cooperation platforms, and sharing cooperation results" at the *Belt and Road International Cooperation Summit Forum* in Indonesia [3]. President Xi's proposal exactly reflects the characteristics of the symbiosis theory, enhancing collective efforts among silk road countries via promoting international investments and collaboration [4]. Symbiosis here, as a term that can be applied in economic domain, refers to the coordinated development of different countries or regions in the economic sector. It is a two-way economy that achieves a symbiosis effect along the way to reach collective development. The symbiosis theory under international production capacity cooperation in the e-commerce sector meets the relative development requirements of the "Belt and Road initiative", mainly through project contracting and direct external methods such as investment and foreign exports will produce higher-quality surpluses transferred abroad. It not only facilitates the gradual internal institutional adjustment and upgrading of the industrial structure but also provides economic growth impetus in various forms.

This article tentatively introduces symbiosis theory research into China-Indonesia production capacity cooperation in the e-commerce sector. Through the study of symbiosis theory and

regional and international economic cooperation, international production capacity cooperation, and the "Belt and Road" initiative compatibility studies, it is found that the characteristics of China-Indonesia production capacity cooperation have sustained constant growth. This article takes China-Indonesia cooperation in the e-commerce field as an example, trying to analyse the symbiosis model and the value chain analysis from the perspective of commodity trade and investment between the two countries–China and Indonesia. With the purpose of investigating the structural features and the investment modes of China-Indonesia e-commerce commodity trade, this paper intends to put forward feasible suggestions to promote the in-depth and effective development of e-commerce capacity building between the two designated countries. Thus, taking the examples of three e-commerce tycoons from China, this article uses a three-level approach, i.e., regional level, inter-firm level, and individual firm level, which incorporates industrial symbiosis [5,6], seeking potential opportunities for business performance improvement for these Chinese e-commerce companies in Indonesia consuming market.

## Literature review

The concept of symbiosis theory, first used in biology research by the German botanist and mycologist Heinrich Anton de Bary [7], was described that dissimilar organisms can live harmoniously in close association with one another for mutual benefits. Scientists have focused on classifying the partnerships and interactions between the bacteria and other microbes [8].

After the 1950s, the dominant thoughts and research paradigms of symbiosis theory gradually penetrated into the fields of philosophy, economics, etc. Following these themes, the theory was widely used in the research of social, humanities, law, and other disciplines. In the late 1990s, mainstream economists in China used symbiosis theory to analyse the characteristics and differences of the small economy in terms of the enterprise system and management system, from the four aspects of symbiosis density, symbiosis interface, symbiosis organization model, and symbiosis behaviour model [9], involving in sociology, management, economics, tourism, and other disciplines. According to the existing definition of the symbiosis theory, the symbiosis relationship includes three components: symbiosis unit, symbiosis model, and symbiosis environment [10], which are similar to the basic characteristics of the comprehensiveness, relevance, and clustering of the international e-commerce industry. The research with the symbiotic relationship as the theme accordingly focuses on the regional e-commerce competition and cooperation relationship, focusing on how to promote the common development of the region by strengthening the industrial linkage to achieve mutual benefit and win-win collaboration. The number of articles on symbiosis has greatly increased since 2007 and China is the country with a number of publications and cases of industrial symbiosis [11,12]. The articles publicised on industrial symbiosis were classified into the literature review, theoretical debates, case studies, and potential industrial symbiosis [13,14]. The methods for quantifying impacts and analysing industrial symbiosis networks were the most widely used in this literature. Out of its symbiosis state, the theory provides a new thinking methodology for the introduction of symbiosis theory into social sciences and economic sectors. As the term "symbiosis" is accepted by more and more scholars, the theory of symbiosis has gradually become a branch in the field of biology, and then evolved into an independent theoretical system.

Industrial symbiosis tools including process integration, mathematical optimization methods, and other models are reviewed and some suggestions are offered on some tools that will enhance the integration of eco-industrial parks [15]. A general mathematical model has been proposed to maximize the total quantity of exchange flows, to maximize the total economic benefice of an industrial park, and to reduce relative environmental pollution, industrial waste treatment cost, and delivery cost [16]. To evaluate the technical potential for industrial

symbiosis, an assess method is proposed allowing for explicitly reflecting current and expected developments at the plant and cluster level [17]. Thus, these e-commerce sectors between China and Indonesia are then subjected to a four-layer analytical investigation within the scope of symbiosis theory, which is separated into regional geographic proximity, national-level policy analysis, supply chain analysis in the economic sector, as well as the organizational-level perspectives.

Current research on industrial symbiosis mainly focuses on four issues: evolution and development, operation carriers, driving mechanisms, and efficiency evaluation of industrial systems [18]. Industrial symbiosis is designated as a subfield of industrial ecology [13], and is defined to include physical exchanges of materials, energy, water, and by-products among diversified clusters of firms [19], with the aim of achieving collective economic, environmental and social gains. The framework conceptualizes symbiosis as a process at two levels: 1. the level of the regional industrial system, and 2. the societal level where the concept and routines of industrial symbiosis diffuse [10].

Social network analysis in symbiosis system focuses on functional entities and how entities are connected. For instance, Gyenge et al. [20] proposes a brand-new marketing strategy model in international e-commerce suppling chain, indicating that enterprises could reach their market goals through effective and delivered communication, and fulfil the integral market needs. This method is discussed in terms of the network density, network centrality, network sub-group, and small world characteristic in analysing the characteristics of industrial symbiosis [18]. Schiller proposed a complexity-derived approach for Social-Material Network Analyses of industrial networks [20]. The social relations of making contact in a matchmaking marketplace is also researched, through investigating a new Australian industrial symbiosis network case and the social relations of inter-organization contact, as a precursor to a waste transfer in an industrial symbiosis matchmaking market [21]. An input-output approach was put forward to form a perfect industrial symbiosis within the production network as a theoretical optimum for industrial symbiosis design where no primary resources are needed from outside and no wastes are discharged outside [22].

The connection between China and Indonesia in terms of its e-commerce collaboration has been shown as a typical example of an industrial symbiosis system. Regional economic cooperation is an important means to optimize the allocation of resources and enhance the competitive advantage among countries in the same region. Deepening policies, mechanisms, and systems have achieved regional economic cooperation and benefit-sharing among ASEAN countries, and market forces have played a leading role. Symbiosis is mainly reflected in the increase in the economic scale and economic speed of both parties, rather than redistributing resources. Therefore, the symbiosis theory is highly compatible with regional economic cooperation.

The Belt and Road Initiative has shown China's overall intention for extensive connection within five domains—policy communication, facility connectivity, unimpeded trade, financial connectivity, and people-to-people networking, and build a mutually beneficial and win-win "community of interests" and a "destiny for common prosperity and development" with countries along with the "land and marine silk road". community". China's "One Belt, One Road" initiative mainly hopes to realize a "community of interests" through international economic cooperation with "One Belt and One Road" countries, and to realize a "community of shared destiny" by means of symbiosis [23].

International economic cooperation refers to the process of mutual adjustment of policies and behaviours between countries to meet the actual or expected economic needs of all parties, and the dynamic process of continuous development along with the world economic practice. The development of symbiosis can increase the economic growth rate and expand the scale of

production, thereby increasing productive employment opportunities. Therefore, the symbiosis theory is also applicable to the development of China's foreign international economic cooperation, and the two have high compatibility. China first proposed the "Belt and Road" initiative at an international conference in Indonesia. Therefore, China and Indonesia play important roles in the Belt and Road initiative, and at the same time have the characteristics of sustainable development in the symbiotic correlations and have a trend of development from asymmetric to symmetric symbiosis in the development stage.

## Research methodology

This article adopts a case study approach to investigate the integral part of the Chinese e-commerce tycoons and their performances, development strategies in the Indonesia market. Case study approach in world e-commerce industry has been extensively applied by existing researches, for instance, Fu et al. [25] studies international cross-border transaction in global e-commerce sector via a case study approach, proposing that asymmetric consortium blockchain is critical in balancing the regulation policies and the profits of the transaction banks in Shenzhen city, China. Case study approach is also applied in empirical studies by Adamczyk [26] to investigate international e-commerce buyers' compulsive and compensative consuming behaviour. This article selects four successful Chinese e-commerce companies as case study, to investigate their performances in the Indonesia market as implementor and practitioners through the perspective of to symmetric symbiosis theory. Later part of the case study consists of a commentary discussion to these Chinese companies reaching a sustainable development goal with a symbiotic correlation between two countries, taking three facets into consideration, namely, the political proximity, supply chain collaboration and the organizational glocalized survival of these studies cases.

Furthermore, public policy analysis approach has also been adopted in the article [27,28]. A number of data and resources are also collected, for instance, national governmental policies from both China and Indonesia, regional collaborative initiatives in Indonesia, documentary resources, government annual releases, as well as the cooperate regulatory and development strategies. These policies are summarized and are subjected to conduct a national policy analysis through the research agenda. The following information then describes the research spectrum and provides a general review of the research targets.

## Research spectrum and case review

In the first quarter of 2020, ASEAN surpassed the EU to become China's largest trading partner. In the context of the deepening of the "Belt and Road" cooperation, Chinese companies are increasingly favouring investment in ASEAN countries. In addition to traditional hot investment destinations such as Singapore and Malaysia, as the local government has gradually increased its investment in recent years, the investment of Chinese companies is also focusing on underdeveloped areas such as Laos and Cambodia. According to China's latest foreign direct investment statistics, Chinese companies regard Indonesia as one of the main destinations for direct investment in ASEAN countries. The amount of investment in Indonesia is second only to the amount of investment in Singapore, which has reached 1.865 billion US dollars, and the investment stock has reached 12.811 billion USD.

Over time the development of e-commerce in Indonesia is so rapid, with the emergence of ideas in maximizing the benefits of internet services that are increasingly mushrooming. Darwis [24] indicated that developed and developing countries are now directly bound to the system of the international economic interdependence and most countries have at least one national asset needed by other countries. Indonesia is the eleventh largest market for e-

commerce with a revenue of US$20 billion in 2019, placing it ahead of Russia and behind Australia. With an increase of 49%, the Indonesian e-commerce market has contributed 16% to the worldwide growth rate in 2019. Nowadays, with the growing number of new e-commerce emerging, making e-commerce competitiveness in Indonesia increasingly increasing. Based on a survey by the Indonesian Internet Service Providers Association (APJII), internet users in Indonesia are close to 197 million users [25].

China and Indonesia established diplomatic relations in 1958. In 1967, as a founding country, Indonesia, together with Singapore, Thailand, the Philippines, and Malaysia, issued the *Declaration on the Establishment of the Association of Southeast Asian Nations* in Thailand, officially proclaiming the establishment of the Association of Southeast Asian Nations. In 2003, China and ASEAN established a strategic cooperative partnership. At present, the total number of overseas Chinese in Indonesia is nearly 10 million, accounting for about 5% of the total population of Indonesia. More than 90% of them have joined Indonesian nationality, making it the country with the largest number of overseas Chinese in the world. People-to-people exchanges between the two countries are relatively frequent, and there is a good foundation for cooperation in the field of small commodity trade, by which the e-commerce industry from China in Indonesia are grounded. Although China-Indonesia e-commerce international capacity cooperation has just started, with the upgrading of China's industry, the two countries have a high degree of coupling in terms of economic structure and can form a symbiotic relationship of sustainable development. As one of the largest e-commerce markets in Southeast Asia, the Indonesian e-commerce market has been developing tremendously in recent years (see Table 1 below). The main reason for this increase is that the shopping behaviours of tech-savvy consumers have changed, and they will be more willing to increase consumer spending for convenience [26]. Due to their sizeable population and the high penetration of internet services and mobile devices, both China and Indonesia have massive digital economies and hence, exciting digital opportunities [27]. The following chart shows the recent investment from the Chinese e-commerce sector into the internet shopping market.

Jingdong, Alibaba, and Tencent, the top three biggest Chinese tech-unicorns, are on the fore frontier in Chinese's shared economy and booming digit advancement. These company's growing momentums are supported by investment from both developed and developing nations, huge Chinese domestic market, as well as all sorts of governmental and organizational

**Table 1. The top four online shopping websites (investments from China) in Indonesia.**

| | |
|---|---|
| Lazada.com | Lazada is a leader in e-commerce in Indonesia and also has operations in Thailand, the Philippines, Singapore, Malaysia, and Malaysia. It is a comprehensive e-commerce platform. Alibaba invested a total of 4 billion U.S. dollars in Lazada. It is already an absolute controlling shareholder. These efforts definitely grind Alibaba's emphasis on the Southeast Asian e-commerce market. |
| Shopee.com | Shopee is Southeast Asia's largest local Internet group e-commerce platform online in South East Asia in 2015. By the end of 2016, it began to recruit Chinese sellers from China, aiming at the South East Asia consuming market. Facilitated and invested by Tencent, it established a team of hundreds of people in Shenzhen to build a complete domestic seller's IT system and service team, to better facilitate the buyers in the region. |
| Bukalapak.com | Achmad Zaky, a graduate from Teknologi Bandung, started the online shopping business and technology start-up in 2011. With his initial investment, he founded the Indonesian C2C market Bukalapak and raised funds from the well-known Indonesian investor Takeshi Ebihara. Today, Bukalapak is busy in expanding the platform and has grown as one of the biggest online shopping platform unicorns in Indonesia. Alibaba invested in the website as one of the stakeholders. |
| JD.id | It was jointly invested by China JD.com and Provident Capital, the largest investment bank in Southeast Asia, and started operations in November 2015. At present, the self-built logistics system is gradually taking advantage of its advantages and growing rapidly. It is a comprehensive online shopping platform. |

supports. The new "Internet Plus" policy is now helping these companies find their way into new fields of investments while fostering the emergence of a new generation of Chinese unicorns [28].

## Research findings

### National-level collaboration policies to foster geographic proximity for an industrial symbiosis system

Political communication is to strengthen intergovernmental cooperation, actively integrate multi-level intergovernmental macro policy exchange mechanisms, deepen the integration of interests, promote political mutual trust, and reach a new consensus on cooperation. Policies supporting the deployment of industrial symbiosis must be enhanced in the future, especially indirect policies, not addressing industrial symbiosis, which are the most important drivers [29]. What role should governmental policy play to facilitate the development of industrial symbiosis is analysed from a dynamic process perspective and policy sciences [30]. Promote rules and measures for policy economic cooperation, reach consensus on solving problems in cooperation, work together pragmatically, and jointly provide policy support for the implementation of large projects. There are several ways to carry out in-depth policy communication: one is to focus on high-level policy communication between the two sides, to negotiate bilateral visits and the "Belt and Road" and other multilateral institutions to actively reach consensus on cooperation. Another institutional effort is to establish key supervising offices to play a role in e-commerce governance. The Economic and Commercial Counsellor's Office from both sides will provide information consultation to Chinese enterprises and assist enterprises in conducting feasibility studies on target projects. At the same time, local policy changes and local public opinion are reported to the higher authorities in a timely manner and actively reflected to the local government and the media to create a favourable public opinion environment to prevent local people from being misled by false information in the social media.

Geographic proximity is said to be a key characteristic in industrial symbiosis. Jensen quantified geographic proximity and provides practitioners with an insight into the movement trends of different political and interest parties [31]. Based on the social-network analysis, regional industrial symbiosis differs methodologically from other resource synergies and is discussed in 6 themes including industrial symbiosis and regional learning, waste minimization assessments, urban industrial symbiosis, and life cycle thinking, energy efficiency, operational carriers, and social aspects [32]. Although geographic proximity is often associated with industrial symbiosis, it is neither necessary nor sufficient—nor is a singular focus on physical resource exchange [33]. Strategic planning is required to optimize the synergies of relocation. In practice, using industrial symbiosis as an approach to commercial operations—using, recovering and redirecting resources for reuse—results in resources remaining in productive use in the economy for longer. This in turn creates business opportunities, reduces demands on the earth's resources, and provides a stepping-stone towards creating a circular economy [34].

### Enhancing e-commerce value chain circle for an industrial symbiosis system

A research agenda in exploring symbiotic supply chains is proposed, from presenting implications both at the strategic/tactical level (self-organized symbiosis, Facilitated symbiosis, and top-down symbiosis) and operational level [35]. In general, a supply chain encompasses every

effort involved in producing and delivering a product from the supplier to the final customer. Furthermore, supply chain management focuses on multiple customer-supplier dyads ultimately spanning from raw material extractors to the final customer [36]. Based on the framework of the supply chain, some scientists derive propositions on the organizational and operational requirements for collaboration in the context of symbiosis networks, related to the supply chain integration and coordination practices [37: 1058–1067]. The suggested e-commerce supply chain can be summarized as follows (see Fig 2):

Sustainable supply chain management must be explicitly extended to include undesired outputs generated along the supply chain, to consider the entire lifecycle of products, and to optimize a product from a total costs' standpoint [38]. The challenges require management to change existing practices and create new production and management systems. In other words, management has to reconceptualize what the supply chain does from a business point of view, who is involved in the supply chain, and how the performance of this supply chain is measured [39]. E-commerce plays a huge role in promoting the integration of China's industries into the global value chain, especially when cross-border e-commerce has developed rapidly in recent years. This provides a new perspective for studying the improvement of global value chains. While we examine the impact of e-commerce on the status of global value chains from the perspective of transaction costs, we can look into the investment model of some Chinese digital giants' performance in the Indonesia market. The role of e-commerce and cross-border e-commerce lies in reducing the transaction costs and increasing marginal gains. The increase in transaction efficiency promotes the formation of a global production network and can improve the status of the national value chain. In this symbiotic circle, countries with higher production efficiency have a higher value chain position in the global production network, while countries with lower production efficiency enjoy the conveniences and higher cost-value commodities from the supply side. That's the reason we proposed further analyse the on social impact of e-commerce and cross-border e-commerce on global value chains. E-commerce, including cross-border e-commerce between China and Indonesia, promotes both country's market status in the global value chain. All kinds of relative sectors, including physical capital and the courier service quality, human capital and overall economic growth, take their advantages with positive impacts on the promotion of global value chain status.

## Organizational-level Chinese e-commerce company on glocalization for an industrial symbiosis system

In Indonesia, Chinese e-commerce companies continue to integrate into the lives of local people. The glocalization practice of these e-commerce companies is of great significance for deepening the global localization theory and exploring how China's e-commerce can penetrate into the global market [glocalization theory refer to [40–44]. In the scope of today's global economy, the integration of the world economy provides many opportunities for enterprises to enter the international stage, but the diversification of culture also makes enterprises face many challenges in the development process. For multinational enterprises, once the problem of cultural diversity is not properly handled, their own business will be affected. Therefore, many multinational enterprises begin to implement the global localization operation strategy in the international market. "Global thinking, Flexible localization" is the basic principle of global localization operation strategy [42]. That is to say, under the guidance of the strategic thinking of global development, combined with the actual situation of foreign markets, to formulate market operation strategies in line with the consumption habits of local customers.

Alibaba Group Holding, together with Tencent and Jingdong, the largest e-commerce companies in China has put globalization ahead of domestic demand and big data/cloud

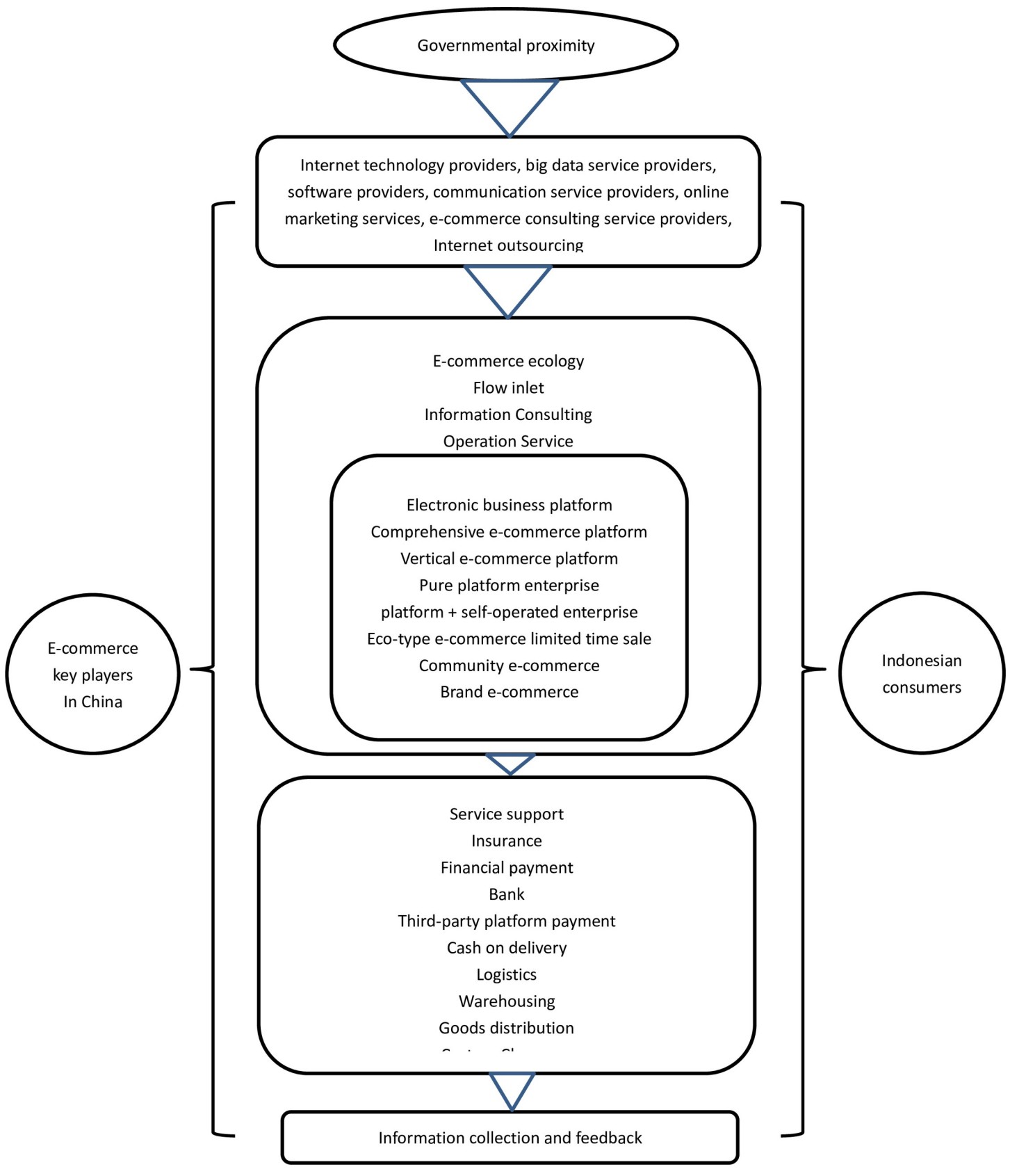

**Fig 2. E-commerce value chain circle between China and Indonesia (summarized by the authors).**

computing in its future strategy, which shows it is inclined to put more effort into exploring overseas markets. This is also in line with the trend for first-tier Chinese companies to explore new markets to support growth, which is unstoppable by external obstruction such as trade protectionism. In this following section, the authors studied three sample enterprises, Tencent, Alibaba and Jingdong from China, which das established joint ventures or sole-owned branches in Indonesia.

## The case study of Tencent

Unicorn investing in overseas markets has become a major move for Tencent to go overseas. The Southeast Asian market has become an important battlefield for Tencent's layout. Tencent and PT Global Mediacom, Indonesia's largest media company, have established a joint venture to develop Indonesia's growing social media market. In the Southeast Asian market, there is a company that has a high mouth-to-mouth reputation, the company named SEA, a gaming and e-commerce company known as the "Little Tencent" in Southeast Asia. SEA was founded in May 2009 and is headquartered in Galaxis, Singapore. It has now developed into a comprehensive platform of digital entertainment, e-commerce and digital financial services, with its mission to improve the ecology of Southeast Asian consumers and small and micro businesses through modern technology. SEA is a member of the "Tencent family", and Tencent is its largest shareholder with a shareholding ratio of approximately 20%. As one of the key Tencent's layouts in the Southeast Asian e-commerce market, SEA established one online shopping store Shopee. In the Southeast Asian e-commerce market, Shopee, backed by Tencent, surpassed Ali's Lazada e-commerce platform and became the Southeast Asian shopping app champion. SEA's businesses also include entertainment Garena and electronic financial services SeaMoney. Calculated from the fourth quarter of 2018, shopee Indonesia's App Store and Google Play app store both ranked first. One of the frontier marketing research institute, aCommerce (see Fig 3 below), did the survey on the shoppers from Shopee and LAZADA on their market preferences [43]:

## The case study of Alibaba

Through the investigation and analysis of Alibaba's series of investments in Southeast Asia, especially the investment of Bukalapak.com, it is found that Alibaba's investment mode and enterprise choice have obvious layout synergy effect, including platform establishment, tech-payment formatting, or logistics system investment. Such advantages lie in the realization of rapid market entry in the short term and the opportunity to develop. On one hand, it can accumulate relevant knowledge and experience for the brand and market qualification required for long-term development. On the other hand, the enterprises invested can achieve the coordination of the whole industry chain in the short term by focusing on vertical development, especially electronic payment with wide application in the Internet era. Because the business scope of the investment enterprises has been covered by many countries, the geographic co-ordination effect can accelerate and promote business cooperation. Alibaba's efforts on Bukalapak.com show that the overseas layout of the e-commerce symbiosis system should be determined with three key sectors: international trading platform, tech-payment forms, as well as the logistics system. Firstly, the investment from Alibaba to Bukalapak.com established a mature platform for connecting the circles of international commodity value chain. Secondly, the financial system, the transaction platform, and the monetary tech-payment system are the most critical areas for the overseas e-commerce layout of these two enterprises. Thirdly, the reality of Indonesia, with thousands of islands, requires these e-commerce tycoons to put the establishment of symbiotic local logistics system as the first strategic consideration. Taking all these factors into

| | LAZADA *Effortless Shopping* | Shopee |
|---|---|---|
| Good reputation | 9.1% | 5.4% |
| Cheaper product price | 13.3% | 22.6% |
| More product selection | 14.2% | 13.2% |
| Authentic products | 0.9% | 1.2% |
| Good customer service | 6.2% | 9.4% |
| Fast delivery | 10.3% | 7.3% |
| Free delivery | 13.2% | 7.2% |
| Easy return policy | 5.5% | 4.1% |
| Easy Navigation on site | 7.5% | 9.0% |
| More payment options | 12.2% | 8.1% |
| Easy communication with sellers | 7.1% | 11.9% |
| More discount | 0.3% | 0.5% |
| | 100% | 100% |

**Fig 3. What shoppers like about their favourite e-marketplace, Cite from (aCommerce 2018).**

consideration, this model shows that it is feasible for Chinese e-commerce companies to create a successful industry symbiosis system in the Indonesian consumer market.

## The case illustration of JD.com

Some scholars review the organizational perspectives found in industrial symbiosis literature using a two-dimensional framework considering the antecedents, consequences, lubricants, and limiters of industrial symbiosis assessed through institutional, network/system, organizational, and individual levels of analysis [44]. Ideas like industrial ecology must become institutionalized with the efforts of academia, industry, and government [45]. The industrial symbiosis operates in two different ways including governmental planning and self-organized ones, and the latter one has proved to be more successful [19]. Chertow focuses on how to uncover and cultivate self-organization industrial symbiosis. Optimization methods applied to the design business model are researched based on societal/managerial objectives, economic objectives, topological objectives, and environmental objectives [46]. With the acceleration of global economic development and the guidance of the "One Belt, One Road" initiatives, many e-commerce platforms are setting their sights on overseas markets and seeking new

development opportunities in Indonesia. In the process of internationalization of major e-commerce platforms, competition is fierce, and only cross-border e-commerce companies with clear strategic paths and obvious advantages can win a place in the global market. At present, more and more companies pay attention to the promotion of brand value, create a successful brand name through distinctive advantages, and embark on the road of brand internationalization. As a well-known domestic e-commerce platform, JD.com has increased its brand value very rapidly in recent years, but it has also faced many difficulties in its development. On the one hand, it has been affected by domestic e-commerce platforms such as "Tmall Taobao" and "Pinduoduo". On the other hand, tech-giants such as "Amazon" are squeezing the market share, and the external environment is changing rapidly. In this context, JD.com chooses brand internationalization, by resorting to other platform-based cross-border e-commerce companies, especially shown in its efforts to establish branch incorporation in the Indonesia market. JD.com is one of the few e-commerce companies from China that established a JD brand in Indonesia, seeking brand internationalization. The strategy and path selection of JD.com sets an example for all Chinese e-commerce companies to go international. The status quo and motivational efforts of JD's brand development to find a foothold for its brand internationalization. E-commerce platform companies should learn from the experience of JD.com in the process of brand internationalization, and also learn from the problems of its brand internationalization, to establish global brand awareness, and to adopt a diversified brand international development strategy suitable for their own development. Maintain and enhance the brand image and enhance the core competitiveness of the enterprise to form a brand advantage.

## Strategic comparison among three Chinese overseas e-business investment models

Cross-border e-commerce technology empowers the world trade industry. The key to the rapid development of e-commerce in today's society lies in the speed of current technology upgrades. Even though the world economy has been largely affected by the devastating impacts of COVID-19, Kitukutha et al. [47] suggest that the rise of e-commerce platforms serves power engines to world economy resilience, to reaching a more sustainable world economy growth momentum. From the Internet, the Internet of Things, cloud computing, big data to artificial intelligence, the iterative update of technologies continues to reshape the development of e-commerce. E-commerce technology provides new possibilities for marketing and terminal sales and gives them unique functions: cross-border e-commerce continues to flatten, corporate information is instantly delivered to global users, provides complex and massive information, and promotes online interaction between merchants and customers. Personalized information push of customers' personal data, endow users with creative power, allow users to re-create their own content consumption (content created by users and social networks generated), etc. The core business of cross-border e-commerce includes commerce, logistics, financial services, data technology, marketing and advertising services, digital media, entertainment, communication platforms, etc.

The business model of China's e-commerce industry is mainly based on the self-operated e-commerce model represented by JD.com and the e-commerce model based on the construction platform represented by Alibaba. As the three largest e-commerce companies in China, the Taobao and Tmall platforms invested by Tencent, Alibaba, and JD.com maintain a continuous competitive relationship in the Chinese market. At the same time, the three companies continued this competitive trend in China-Indonesia cross-border e-commerce investment. From a strategic perspective, the technical competitiveness of Alibaba, Tencent and JD.com in

cross-border e-commerce has greatly promoted the development of the Indonesian e-commerce industry and greatly affected the pattern of the Indonesian e-commerce industry. Chinese cross-border e-commerce companies in the Indonesian market can increase market share through technological innovation strategies, continue to increase investment in advanced e-commerce technologies, and through close cooperation with local companies, actively embedding in the local market and gaining competitive advantages. By integrating the perspective of symbiosis theory, the author compares the development strategies of the three companies in the Indonesian market and proposes that Chinese cross-border e-commerce companies should adopt an embedded survival model. The following is a comparison of the concept of "embedded survival": Achieving that China-Indonesia cross-border e-commerce has moved from an asymmetric symbiosis state to a symmetric symbiosis state, promoting the healthy and vigorous development of the digital economy of the two countries.

First, Chinese e-commerce companies need to pay full attention to technological capability innovation and the construction of cross-border e-commerce value chain systems, actively adopt and integrate into Indonesian local organizations and cultural circles, build a China-Indonesia cross-border e-commerce ecosystem, and straighten out the two countries The integration of cross-border e-commerce ecological value chain. Commercial transactions involve the exchange of value across organizational or individual boundaries (such as money) in exchange for product or service returns. E-commerce has revolutionized traditional commerce and broke the boundaries of time and space. Cross-border e-commerce conducts non-face-to-face transactions through the Internet or mobile devices by building business platforms with digital functions, introducing digital technology. Through the integration of the three networks, the enrichment of products and the improvement of service quality, Tencent strengthens the dependence of local customers. The Internet of Things technology has great potential in the field of e-commerce and is very effective in enhancing its automation. The Internet of Things technology will greatly promote Tencent's operations and management in Indonesia and improve the shopping experience of customers and Internet users.

Second, the embedded survival strategy of Chinese e-commerce companies must build a cross-border e-commerce platform with a global localized vision. Alibaba Group has become one of the most active and powerful companies in China's Internet technology research and development. The establishment of Alibaba's cross-border e-commerce platform has pushed the e-commerce revolution into a new journey. The most important value of a cross-border e-commerce platform is that it can quickly attract global users regardless of geographic location and break space constraints. This has greatly helped small and micro-enterprises to open up the market and made it unnecessary to bear huge amounts of capital and excessive organizational resource costs. At the same time, it also promoted innovative e-commerce platforms to continue to promote the leapfrog development of cross-border e-commerce and expand overseas investment opportunities. However, even though Alibaba has begun to increase its investment in logistics service capabilities when facing logistics problems in the Indonesian market, it has not paid enough attention to the difficulties in the construction of local logistics services in Indonesia.

Third, the embedded survival strategy of cross-border e-commerce companies should also pay full attention to the construction and improvement of multinational brand competitiveness. Successful brand management can not only enable the company to maintain a higher market share and profit and increase its value, but also build barriers to competition and prevent competitors from participating in the competition. JD's brand competitiveness will greatly affect local customer loyalty and recommendation rate. Although JD's technical competitiveness is significantly lower than that of Alibaba, and there are higher cost barriers, the self-operated logistics system has greatly helped JD's competitiveness and made JD's

adaptability more prominent in the Indonesian market. At the same time, the establishment of a JD branch in Indonesia will provide good embedded conditions for the development and growth of the JD brand in the Indonesian market. By embedding a large number of offline resources (brands, channels, customers) with Indonesia's traditional retail companies, JD's Indonesia branch is relatively more dynamic and competitive, which is mainly reflected in its good brand influence, complete logistics system, and more Stable supply channels and perfect after-sales service capabilities have achieved good industrial symbiosis effects.

## Conclusion and discussion

Through analysing the investment modes, overall development situation, and the survival status of representative e-commerce tycoons in China-Indonesia e-commerce markets, this article combining data analysis, government public policy analysis and case study, has described how symbiosis theory as a modern cooperation model can be used to promote e-commerce collaboration between China and Indonesia, basically from the 4 perspectives of regional geographic proximity, national-level policy analysis, supply chain analysis in the economic sector, and the organizational-level perspectives. More specifically, the national-level policy can be applied to foster geographic proximity in symbiosis system, stressing that high-level policy communication contributes to reaching consensus, and the key supervising offices facilitate e-commerce governance. Besides, enhancing e-commerce supply chain circle is of great importance to symbiosis system. Moreover, applying organizational-level Chinese e-commerce companies of Alibaba and Jingdong, as successful enterprise globalization cases, to the globalized strategy practices in the international market, in which this article specially focuses on Indonesia market. It suggests that, currently, Chinese-Indonesia cooperation is shifting from the original asymmetric symbiosis to the symmetric symbiosis, which indicates a more balanced and equal collaboration between two nations in the e-commerce sector.

This paper connects symbiosis theory with Chinese overseas investment in e-commerce. It investigates the evolution of the symbiosis theory and its application in e-commerce sector. It is suggested that geopolitical proximity can be viewed as a development premise for ecommerce collaboration between China and Indonesia. Furthermore, it analyses the changing landscape of value chain system between China and Indonesia. It is advocated that a glocalized survival strategy can be adopted. Shortcomings of this article mainly provides the positive aspects of Chinese-Indonesia e-commerce symbiosis mechanism, and studies the impact of e-commerce development in different countries. limitations of the article lie in the fact that there are negative impacts, such as obstacles in the process of cooperation, the negative impacts of symbiosis. These limitations are yet to be discussed in detail. Academia are suggested to further future investigations on the downsides of symbiosis theory in practice, as well as consider how to overcome the existing obstacles and blockings, and how to foster a more effective e-commerce collaboration between China and Indonesia.

## Author Contributions

**Conceptualization:** Jinsheng (Jason) Zhu, Xianchun Zhang.

**Data curation:** Jinsheng (Jason) Zhu.

**Formal analysis:** Jinsheng (Jason) Zhu.

**Funding acquisition:** Jinsheng (Jason) Zhu.

**Investigation:** Jinsheng (Jason) Zhu, Weidian Lan, Xianchun Zhang.

**Methodology:** Jinsheng (Jason) Zhu, Xianchun Zhang.

**Project administration:** Jinsheng (Jason) Zhu, Xianchun Zhang.

**Resources:** Jinsheng (Jason) Zhu.

**Software:** Jinsheng (Jason) Zhu.

**Supervision:** Jinsheng (Jason) Zhu, Xianchun Zhang.

**Validation:** Jinsheng (Jason) Zhu, Weidian Lan.

**Visualization:** Jinsheng (Jason) Zhu, Weidian Lan.

**Writing – original draft:** Jinsheng (Jason) Zhu.

**Writing – review & editing:** Jinsheng (Jason) Zhu.

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
