## [Decision Letter · Decision Letter 0]

16 Jul 2021

PONE-D-21-18757

Geographic proximity, Supply Chain and Organizational Glocalized Survival: A Perspective of Symbiosis Theory on Chinese E-commerce Investment in Indonesia

PLOS ONE

Dear Dr. Jinsheng Jason Zhu,

Thank you for submitting your manuscript to PLOS ONE. After careful consideration, we feel that it has merit but does not fully meet PLOS ONE’s publication criteria as it currently stands. Therefore, we invite you to submit a revised version of the manuscript that addresses the points raised during the review process.

We look forward to receiving your revised manuscript.

Kind regards,

László Vasa, PhD

Academic Editor

PLOS ONE

Journal Requirements:

2. Please include your tables as part of your main manuscript and remove the individual files. Please note that supplementary tables (should remain/ be uploaded) as separate "supporting information" files

3. Please ensure that you refer to Figures 2 and 3 in your text as, if accepted, production will need this reference to link the reader to the figure.

Reviewers' comments:

Reviewer's Responses to Questions

**Comments to the Author**

1. Is the manuscript technically sound, and do the data support the conclusions?

Reviewer #1: Yes

Reviewer #2: Partly

2. Has the statistical analysis been performed appropriately and rigorously? 

Reviewer #1: Yes

Reviewer #2: N/A

3. Have the authors made all data underlying the findings in their manuscript fully available?

Reviewer #1: Yes

Reviewer #2: Yes

4. Is the manuscript presented in an intelligible fashion and written in standard English?

Reviewer #1: Yes

Reviewer #2: Yes

5. Review Comments to the Author

Reviewer #1: Comments and Suggestions for Authors

Thank you for the opportunity to read the study paper. Topic is very interesting but at this stage of this report I cannot judge how important it is. Some additional data and resource citation should support this topic (in introduction). The topicality is seemingly supported by the fact of glocalization concept but e-commerce is not fully topical. Although any surviving topic is likely to be trendy and interesting to readers of the journal.

Title: it is rather too complex. In the title we get or suspect five topics at once, Geographic proximity, glocalization, survival economy, symbiosis theory, E-commerce investment in Indonesia. These make too wide scope to address the topic. It needs to be more clearer. My suggestion is: Geographic proximity, Supply Chain and Organizational Glocalized Survival perspectives in Indonesia via e-commerce investments by China. (or something like that)

Editing and formation of the article: The article submitted does not fully meet PLOS ONE’s publication criteria in its current state. The article does not use numbered template file and in the text do not use parentheses instead of brackets, an so on. It should be matched together. Author is not native English speaker so she is advised to do thorough English language check.

Abstract and introduction is not well-structured and clear at the present form. It should focus on the importance of the topic and relevant (or main) issue more. The research goals and gaps, context and purpose are blurry in current stage. (In addition in the introduction some new literature sources should be cited especially in the following areas.)

a, symmetric and asymmetric, what's more symmetric symbiosis theories (in economy)

b, main scientific issues in the field, scientific uncertainties and gaps between the theories

c, I suggest incorporating the newest debates of prestigious international journals with foreign authors also.

“Highlights” should be revised as goals or issues and incorporated as an integral part of the introductory section. It should be highlighted how these study contributes to the development of scientific knowledge.

The literature review. The literature review section is modest and I would request you to improve Literature section. You should make comprehensive comparisons and extensively cite previously published articles in Plos One or other international journals.

Research spectrum and research methodology. The methodology is not well defined, but it seems to secondary analysis mainly. Some more specification is needed and each of the statements, statistical data resource need to be cited. The research design must be explained including analyses and test performed, and arguments that sustained them suitability should be provided.

Research findings – These sections need to be reviewed and linked more clearly to research issues and methodology. Please include the limitations of statements and explain it how they are based on the data.

Case studies – Please also include the final conclusions.

Conclusion and discussion. Please clarify what are the main conclusions, what are the hierarchy of them. In the last section of the paper, the conclusion needs to include the paper theoretical contribution (practical/managerial contribution) limitations and future research perspectives.

The paper has a good potential but in this stage request some more work.

Reviewer #2: The paper focuses on a relatively up-to-date topic; the need to know more about e-commerce issues increases. So, the paper's topic can be evaluated as actual and can be of interest to PLOS ONE readers.

The abstract is written well but not perfect; the methodology used in the research should be highlighted in one sentence. Also, among the keywords, one should indicate the methodology.

The paper starts with a "Highlights" chapter which is not necessary; it should be integrated into the introduction.

The introduction is acceptable.

The literature review should be extended with few more latest sources published in Q category journals (e.g., https://doi.org/10.3390/jtaer16040064 and https://doi.org/10.23762/fSo_Vol9_no2_3).

Regarding the methodology chapter, we cannot find any description of the methodology used. The research spectrum is clear, but how the author got the results based on what methods?

The research findings chapter is structured along with some statements which sound like hypotheses. Case studies follow these. The results seem to be clear, but the main question and concern is how the author executed the research; with the help of what methodology?

Case studies are well written; however, I am not sure these are needed among the results. BTW, the methodology chapter is not written anything about these case studies (=case study itself is one of the methods).

I recommend rethinking the structure of the results, extending the literature review and, rewriting the methodology chapter. Then, after improvements and new reviews, the paper could be eligible for publication.

6. PLOS authors have the option to publish the peer review history of their article (what does this mean?). If published, this will include your full peer review and any attached files.

Reviewer #1: No

Reviewer #2: No

---

## [Author Response · Author response to Decision Letter 0]

31 Jul 2021

Dear Distinguished Editor of PLOS ONE, 

Thank you very much for your encouraging email to request a major revision of the paper coded as PONE-D-21-18757. I am indeed grateful for your editorial decision to give us an opportunity to resubmit. 

I have closely examined the reviewers’ comments and your own suggestions, and then listed here the revisions I have made as follows with red color bold font:

Reviewer's Responses to Questions

Comments to the Author

1. Is the manuscript technically sound, and do the data support the conclusions?

Reviewer #1: Yes

Reviewer #2: Partly

Response to Reviewers: Thank you for your positive comments on the paper. I will be revising this paper according to your most valued review comments below to fit with the criteria of this prestige journal. 

2. Has the statistical analysis been performed appropriately and rigorously? 

Reviewer #1: Yes

Reviewer #2: N/A

Response to Reviewers: I will continue working on the paper with proper statistical analysis. 

3. Have the authors made all data underlying the findings in their manuscript fully available?

Reviewer #1: Yes

Reviewer #2: Yes

Response to Reviewers: Thank you so much, dear reviewers. If one day I can be a reviewer for PLOS ONE, I will also keep high standard to meet the requirement of the journal, like you.

4. Is the manuscript presented in an intelligible fashion and written in standard English?

Reviewer #1: Yes

Reviewer #2: Yes

Response to Reviewers: Thank you indeed. 

Review Comments to the Author

Response to Reviewers: I will attach the statement of sole submission to your journal, conflict of interest statement, as well as research ethics statement in the resubmission process. 

Response to Reviewer # 1

Reviewer #1: Comments and Suggestions for Authors

Thank you for the opportunity to read the study paper. Topic is very interesting but at this stage of this report I cannot judge how important it is. Some additional data and resource citation should support this topic (in introduction). The topicality is seemingly supported by the fact of glocalization concept but e-commerce is not fully topical. Although any surviving topic is likely to be trendy and interesting to readers of the journal.

Title: it is rather too complex. In the title we get or suspect five topics at once, Geographic proximity, glocalization, survival economy, symbiosis theory, E-commerce investment in Indonesia. These make too wide scope to address the topic. It needs to be clearer. My suggestion is: Geographic proximity, Supply Chain and Organizational Glocalized Survival perspectives in Indonesia via e-commerce investments by China. (or something like that)

Response to Reviewers: This is really a great comment! For me, I feel that my original title is too long and not powerful enough to attract PLOS ONE readers. I will take this advice to revise the topic. Shorter one will be more powerful, attractive and convincing. How about this: “Geographic proximity, Supply Chain and Organizational Glocalized Survival: China’s e-commerce investments in Indonesia” ? I have copy paste this title in the revised manuscript and will attach it in the paper submission system.

Editing and formation of the article: The article submitted does not fully meet PLOS ONE’s publication criteria in its current state. The article does not use numbered template file and in the text do not use parentheses instead of brackets, an so on. It should be matched together. Author is not native English speaker so she is advised to do thorough English language check.

Response to Reviewers: I firstly downloaded the PLOS ONE formatting sample and revised the format of the paper, following the criteria of the journal. Learning from the template, 

Abstract and introduction is not well-structured and clear at the present form. It should focus on the importance of the topic and relevant (or main) issue more. The research goals and gaps, context and purpose are blurry in current stage. (In addition in the introduction some new literature sources should be cited especially in the following areas.)

a, symmetric and asymmetric, what's more symmetric symbiosis theories (in economy)

b, main scientific issues in the field, scientific uncertainties and gaps between the theories

c, I suggest incorporating the newest debates of prestigious international journals with foreign authors also.“Highlights” should be revised as goals or issues and incorporated as an integral part of the introductory section. It should be highlighted how these study contributes to the development of scientific knowledge.

Response to Reviewers: Thank you so much for your kind suggestion, I think in the original version of the manuscript, the highlights section indicates the article’s academic contribution to the existing literatures. Thus, adopting your comments, I have incorporated this highlights section as an integral part in the conclusion section in page 27. 

The literature review. The literature review section is modest, and I would request you to improve Literature section. You should make comprehensive comparisons and extensively cite previously published articles in Plos One or other international journals.

Response to Reviewers: I have included citations from [48] and [49] from the prestige journal of PLOS ONE, including Gulc’s idea on courier service quality and Anvari et al. on the idea of impact of e-commerce and its R&D sections. These citations increase the literatures in the e-commerce sector and the quality-based studies. 

Research spectrum and research methodology. The methodology is not well defined, but it seems to secondary analysis mainly. Some more specification is needed and each of the statements, statistical data resource need to be cited. The research design must be explained including analyses and test performed, and arguments that sustained them suitability should be provided.

Response to Reviewers: In this section, I divided the research spectrum and research methodology into two part, separating research methodology as separate part, including the introduction of main research approach of the article, for instance, the case study approach. In the explanatory part of methodology, I introduced the methods adopted in the article. Better description of research methodology serves as better explanatory chapter for a clearer research spectrum. 

Research findings – These sections need to be reviewed and linked more clearly to research issues and methodology. Please include the limitations of statements and explain it how they are based on the data.

Response to Reviewers: I tried to enhance the correlations between research findings with the research methodology by increasing two paragraphs in the research methodology section, to clearly state the research methodology of case study and public policy analysis. 

Case studies – Please also include the final conclusions.

Response to Reviewers: The final conclusions in page 27, the section of introducing case study approach and governmental resources analysis has been added to the text for serving as conclusive remarks of all the related sections in the article. 

Conclusion and discussion. Please clarify what are the main conclusions, what are the hierarchy of them. In the last section of the paper, the conclusion needs to include the paper theoretical contribution (practical/managerial contribution) limitations and future research perspectives.

Response to Reviewers: Conclusion part has been amended with an insertion of one paragraph about its theoretical contribution to the existing literature in page 28. Limitations and future research suggestions are also included in the latter part of the conclusion part. 

The paper has a good potential but in this stage request some more work.

Response to Reviewers: Thank you indeed for your encouraging comments and precise amendment suggestions on the article. I am indeed very grateful for your advices. I am cordially revising the article exactly based on your most valuable comments. Thank you very much for giving me this chance to meet with an insightful reviewer with sharp academic wisdom. 

Response to Reviewer # 2

Reviewer #2: The paper focuses on a relatively up-to-date topic; the need to know more about e-commerce issues increases. So, the paper's topic can be evaluated as actual and can be of interest to PLOS ONE readers.

Response to Reviewers: Thank you very much for your encouraging comments. I am grateful to encounter with you and your brilliant wisdom. 

The abstract is written well but not perfect; the methodology used in the research should be highlighted in one sentence. Also, among the keywords, one should indicate the methodology.

Response to Reviewers: In the abstract part, I have revised the sentence to be more precise and neatly, mentioning the research methodology in one sentence. Furthermore, I have included case study and public policy analysis in the keywords section. 

The paper starts with a "Highlights" chapter which is not necessary; it should be integrated into the introduction.

Response to Reviewers: I have integrated the highlights chapter in the conclusion part, since the five highlight sentences are originally summarizing the practical and theoretical contributions to the existing academic literatures. This part better suits the highlights in the first submitted manuscript. Hope you like this idea. 

The introduction is acceptable.

Response to Reviewers: Thank you very much for your encouragement. 

The literature review should be extended with few more latest sources published in Q category journals (e.g., https://doi.org/10.3390/jtaer16040064 and https://doi.org/10.23762/fSo_Vol9_no2_3).

Regarding the methodology chapter, we cannot find any description of the methodology used. The research spectrum is clear, but how the author got the results based on what methods? 

Response to Reviewers: In this part, I have revised the the research spectrum and research methodology into two part, separating research methodology as separate part, including the introduction of main research approach of the article, for instance, the case study approach and public policy analysis. In the explanatory part of methodology, I introduced the methods adopted in the article. 

The research findings chapter is structured along with some statements which sound like hypotheses. Case studies follow these. The results seem to be clear, but the main question and concern is how the author executed the research; with the help of what methodology? 

Response to Reviewers: Your queries are of great importance for me to revise the methods and spectrum part, for a better description of research methodology serves as better explanatory chapter for a clearer research spectrum. It also helps in setting a reasonable methodological foundation for the research findings and the case studies section. 

Case studies are well written; however, I am not sure these are needed among the results. BTW, the methodology chapter is not written anything about these case studies (=case study itself is one of the methods).

Response to Reviewers: With an aforementioned revision to the research methodology revision, the whole structure of the paper became more solid and formed with academic logical sequence. Your review to the structural settings of this article largely enhances the academic foundation. I am indeed grateful for this. 

I recommend rethinking the structure of the results, extending the literature review and, rewriting the methodology chapter. Then, after improvements and new reviews, the paper could be eligible for publication.

Response to Reviewers: Thank you very much. I have attained to your most valuable comments accordingly. Your reviews deserve my sincere gratitude and appreciation.

---

## [Decision Letter · Decision Letter 1]

17 Aug 2021

Geographic proximity, supply chain and organizational localized survival: China's e-commerce investment in indonesia

PONE-D-21-18757R1

Dear Dr. Zhang,

We’re pleased to inform you that your manuscript has been judged scientifically suitable for publication and will be formally accepted for publication once it meets all outstanding technical requirements.

Kind regards,

László Vasa, PhD

Academic Editor

PLOS ONE

Additional Editor Comments (optional):

Reviewers' comments:

Reviewer's Responses to Questions

**Comments to the Author**

1. If the authors have adequately addressed your comments raised in a previous round of review and you feel that this manuscript is now acceptable for publication, you may indicate that here to bypass the “Comments to the Author” section, enter your conflict of interest statement in the “Confidential to Editor” section, and submit your "Accept" recommendation.

Reviewer #1: All comments have been addressed

Reviewer #2: All comments have been addressed

2. Is the manuscript technically sound, and do the data support the conclusions?

Reviewer #1: Yes

Reviewer #2: Yes

3. Has the statistical analysis been performed appropriately and rigorously? 

Reviewer #1: Yes

Reviewer #2: Yes

4. Have the authors made all data underlying the findings in their manuscript fully available?

Reviewer #1: Yes

Reviewer #2: Yes

5. Is the manuscript presented in an intelligible fashion and written in standard English?

Reviewer #1: Yes

Reviewer #2: Yes

6. Review Comments to the Author

Reviewer #1: The authors have improved the paper as recommended, so I accept it fully for publication. Thanks for the opportunity to read the study paper.

Reviewer #2: The authors accepted the reviewers' recommendations and improved their paper well. Abstract clarified, more literature added, research methodology chapter added, conclusions rewritten. After these modifications, I find it eligible for publishing in the journal.

7. PLOS authors have the option to publish the peer review history of their article (what does this mean?). If published, this will include your full peer review and any attached files.

Reviewer #1: No

Reviewer #2: No

---

## [Editor Report · Acceptance letter]

27 Aug 2021

PONE-D-21-18757R1 

Geographic proximity, Supply Chain and Organizational Glocalized Survival: China’s e-commerce investments in Indonesia 

Dear Dr. Zhang:

I'm pleased to inform you that your manuscript has been deemed suitable for publication in PLOS ONE. Congratulations! Your manuscript is now with our production department. 

Kind regards, 

on behalf of

Prof. Dr. László Vasa 

Academic Editor

PLOS ONE